# Separating radiative forcing by aerosol–cloud interactions and rapid cloud adjustments in the ECHAM–HAMMOZ aerosol–climate model using the method of partial radiative perturbations

Johannes Mülmenstädt[1], Edward Gryspeerdt[2], Marc Salzmann[1], Po–Lun Ma[3], Sudhakar Dipu[1], and Johannes Quaas[1]

[1]Universität Leipzig, Leipzig, Germany
[2]Space and Atmospheric Physics Group, Imperial College London, London, United Kingdom
[3]Pacific Northwest National Laboratory, Richland, Washington, USA

**Correspondence:** Johannes Mülmenstädt (johannes.muelmenstaedt@uni-leipzig.de)

**Abstract.** Using the method of offline radiative transfer modelling within the partial radiative perturbations (PRP) approach, the effective radiative forcing by aerosol–cloud interactions ($ERF_{aci}$) in the ECHAM–HAMMOZ aerosol climate model is decomposed into a radiative forcing by anthropogenic cloud droplet number change and adjustments of the liquid water path and cloud fraction. The simulated radiative forcing by anthropogenic cloud droplet number change and liquid water path adjustment are of approximately equal magnitude at $-0.52\,\mathrm{W\,m^{-2}}$ and $-0.53\,\mathrm{W\,m^{-2}}$, respectively, while the cloud fraction adjustment is somewhat weaker at $-0.31\,\mathrm{W\,m^{-2}}$ (constituting 38%, 39%, and 23% of the total $ERF_{aci}$, respectively); geographically, all three $ERF_{aci}$ components in the simulation peak over China, the subtropical eastern ocean boundaries, the northern Atlantic and Pacific oceans, Europe, and eastern North America (in order of prominence). Spatial correlations indicate that the temporal-mean liquid water path adjustment is proportional to the temporal-mean radiative forcing, while the relationship between cloud fraction adjustment and radiative forcing is less direct. While the estimate of warm-cloud $ERF_{aci}$ is relatively insensitive to the treatment of ice and mixed-phase cloud overlying warm cloud, there are indications that more restrictive treatments of ice in the column result in a low bias in the estimated magnitude of the liquid water path adjustment and a high bias in the estimated magnitude of the droplet number forcing. Since the present work is the first PRP decomposition of the aerosol effective radiative forcing into radiative forcing and rapid cloud adjustments, idealized experiments are conducted to provide evidence that the PRP results are accurate. The experiments show that using low-frequency (daily or monthly) time-averaged model output of the cloud property fields underestimates the ERF, but three-hourly mean output is sufficiently frequent.

## 1 Introduction

Following Boucher et al. (2014), it has become common to distinguish between radiative forcing (RF) by aerosol–cloud interactions (ACI) and rapid adjustments to this radiative forcing, with the sum of the forcing ($RF_{aci}$) and the rapid adjustments

denoted as the effective radiative forcing ($ERF_{aci}$). In liquid-water clouds, $RF_{aci}$ arises from the increased availability of cloud condensation nuclei (CCN) in a polluted atmosphere leading to higher droplet number $N_d$ and smaller effective radius $r_e$ at constant cloud liquid water path $\mathcal{L}$ (Twomey, 1977). The atmosphere responds to the higher-$N_d$, lower-$r_e$ clouds by various processes occurring on short timescales, leading to adjustments in other cloud properties, including $\mathcal{L}$ and cloud vertical and horizontal geometric extent.

Physically, the most important adjustment mechanisms are suppression of precipitation formation (Albrecht, 1989) and enhanced cloud-edge dry-air entrainment and droplet evaporation in the smaller-droplet clouds (Ackerman et al., 2004). The former mechanism, in isolation, would lead to an increase in cloud condensate and cloud fraction, and thus a negative adjustment to the radiative forcing; the latter, in isolation, would lead to a decrease in cloud condensate and thus a positive adjustment to the radiative forcing. Since no mechanism occurs in isolation in a coupled system (Stevens and Feingold, 2009), the question of whether the net adjustment is positive or negative is both difficult and unresolved (e.g., Mülmenstädt and Feingold, 2018; Gryspeerdt et al., 2019a).

One method by which aerosol effective radiative forcing is estimated is to calculate the difference in top-of-atmosphere (TOA) radiative fluxes in general-circulation models (GCMs) between runs with fixed sea-surface temperature and present-day or preindustrial aerosol concentrations or emissions (e.g., Lohmann et al., 2010; Forster et al., 2016). This requires a GCM that includes the relevant aerosol–radiation and aerosol–cloud interaction mechanisms. By representing the optical properties of aerosol in the radiative transfer, the radiative forcing by aerosol–radiation interactions ($RF_{ari}$) can be estimated; by representing the $N_d$ dependence on aerosol activation during cloud formation, $RF_{aci}$ can be estimated; and by representing precipitation suppression and enhanced evaporation in smaller-$r_e$ clouds, the adjustments to $RF_{aci}$ can be estimated. (We exclude ice and mixed-phase cloud processes, which introduce further complications, from this discussion.) These processes occur on scales far below the resolved scale, so their representation in the GCM requires parameterization. Thus, the model is only imperfectly (if at all) aware of subgrid-scale variability in the process rates and feedbacks between the processes; relies on imperfect base-state cloud properties (e.g., Penner et al., 2006); and only considers effects that are amenable to parameterization, meaning that precipitation suppression is included in many models but enhanced evaporation is not (e.g., Salzmann et al., 2010; Michibata et al., 2016; Zhou and Penner, 2017). Based on these considerations, a prevalent view is that, from the standpoint of achieving GCM fidelity, ACI are more difficult than aerosol–radiation interactions, and ACI adjustments are more difficult than the ACI forcing. On top of this, the usual concerns about parametric uncertainty apply, so that the overall uncertainty on GCM estimates of $ERF_{aci}$ is large (Boucher et al., 2014). Nevertheless, the ability of GCMs to produce a global estimate will assure their star will continue to shine brightly in the firmament of ERF estimation methods until competing methods overcome their own significant drawbacks.

In general, one might argue that knowing the uncertainty on each term in a sum is a good first step towards attacking the uncertainty on the total; certainly, this is consistent with the GCM paradigm of building up the total forcing from parameterizations for each of the contributing processes, even if it is less applicable to "top-down" estimates from the historical evolution of the climate system. Thus, we write the effective radiative forcing by aerosol as

$$F = F_{ari} + F_{N_d} + F_{\mathcal{L}} + F_{f_c}, \tag{1}$$

where $F_{ari}$ is the RF$_{ari}$, $F_{N_d}$ is the RF$_{aci}$ due to the increase in $N_d$, and $F_\mathcal{L}$ and $F_{f_c}$ are the $\mathcal{L}$ and cloud fraction ($f_c$) adjustments to the RF$_{ari}$ and RF$_{aci}$. Other adjustments, e.g., due to rapid changes in land surface temperatures or atmospheric temperature and humidity profiles, have been estimated as small in a previous study (Heyn et al., 2017). Each of these terms maps fairly well onto a parameterization in the GCM: RF$_{ari}$ is parameterized in the radiative transfer, $F_{N_d}$ is parameterized in a droplet activation

scheme, the ACI part of $F_\mathcal{L}$ is parameterized in the precipitation microphysics (and, if enhanced evaporation becomes tractable in the future, that component will presumably be parameterized in the turbulence scheme; e.g., Guo et al., 2011; Neubauer et al., 2014), and $F_{f_c}$ is parameterized in the cloud cover scheme (although in our model the response to the perturbation is indirect, via relative humidity changes subsequent to precipitation rate changes); the only component that emerges from the model dynamics rather than from an explicit parameterization is the adjustments of temperature and moisture profiles that

entail further adjustments to aerosol–cloud interactions and to aerosol–radiation interactions (formerly known as "semi-direct effect"); both of these terms are small (Heyn et al., 2017; Stjern et al., 2017). A natural first step would then be to ask how large each of these terms is, and a natural second step would be to ask how much uncertainty each contributes to the total. One of the benefits of such a decomposition would be that it would provide a more solid footing for—or falsify—the notion that models agree fairly well on the "simpler" problem of RF$_{aci}$ and not well at all on the "harder" problem of the adjustments.

However, performing the decomposition is quite difficult in practice. Ghan (2013) addresses the issue of separating $F_{ari}$ from $F_{N_d} + F_\mathcal{L} + F_{f_c}$ using the model's intrinsic knowledge of the anthropogenic perturbations. It would be desirable to separate the three ERF$_{aci}$ components analogously, using the intrinsic model knowledge of the time-varying, three-dimensional aerosol perturbation and resulting perturbation of cloud properties. However, methods that attempt to do so need to contend with the problem that ERF$_{aci}$ is diagnosed from two separate runs that cannot easily share fields online, so that double radiation

calls as in Ghan (2013) are not feasible; furthermore, even if double radiation calls could be made, this would be of little help in estimating adjustments, which, by definition, do not respond to the aerosol perturbations instantaneously. One method to estimate ERF$_{aci}$ components is the approximate partial radiative perturbation (APRP) decomposition (Taylor et al., 2007; Zelinka et al., 2014). APRP decomposes the cloud property changes into changes in area fraction, cloud albedo, and cloud absorption; the change in area fraction maps well onto the $f_c$ adjustment in the forcing–adjustment framework, but the APRP

cloud albedo change includes both the effect of the anthropogenic $N_d$ change and the $\mathcal{L}$ adjustment. Another method is to deactivate the parameterized precipitation suppression (and, if models include a parameterization of the enhanced evaporation, deactivate that as well); however, the model with parameterized adjustments will produce a different climate than the model without (Penner et al., 2006). Due to the complications arising in methods that directly use the model state, less direct methods have been developed that idealize the cloud field as a globally homogeneous single layer (Ghan et al., 2016) or use similar

regression-based statistical techniques as satellite studies (Gryspeerdt et al., 2019b).

  In this work, we apply the method of partial radiative perturbations (PRP; Wetherald and Manabe, 1988; Colman and McAvaney, 1997; Colman, 2003; Klocke et al., 2013) to the ERF decomposition problem. PRP falls in the category of methods that directly use the intrinsic model knowledge of the time-varying, three-dimensional aerosol perturbation and resulting perturbation of cloud properties. The starting point for PRP is a perturbed and an unperturbed model run. One then introduces the fields

of the perturbed run into the unperturbed run, one at a time, and reruns the radiative transfer scheme on the "partially perturbed"

state to derive the resulting change in radiative fluxes. In our application, the two runs are simulations with an atmospheric GCM with prescribed climatological, seasonally varying sea surface temperature (SST) and sea ice cover (SIC) distributions with present-day and preindustrial aerosol emissions, nudged to present-day large-scale upper-level winds to reduce the internal variability without overconstraining the behavior of lower-tropospheric warm cloud and allow significant changes in cloud property to emerge after a shorter integration time than would otherwise be required (e.g., Kooperman et al., 2012; Zhang et al., 2014). The perturbed fields are $N_d$, $\mathcal{L}$, and $f_c$; the corresponding changes in radiative fluxes are $F_{N_d}$, $F_{\mathcal{L}}$, and $F_{f_c}$.

In Section 2, we describe the PRP method and ECHAM–HAMMOZ model in detail; in Section 3, we use PRP to estimate the ERF components in the ECHAM–HAMMOZ model and determine whether the adjustments are a simple proportional response to the forcing.

## 2   Methods

We first give a brief formal description of the PRP method; we then describe the model configurations to which we will apply the method.

### 2.1   Partial radiative perturbations

We denote the shortwave TOA flux as $Q$ and the longwave flux as $R$ (all-sky, positive downward in both cases). For the purposes of this analysis, the radiative flux in each spectral range is considered a function of the cloud properties $N_d$, $\mathcal{L}$ (or the vertically resolved analogue $q_l$), and $f_c$. The dependence of the fluxes on other climate state variables – water vapor mixing ratio $q$; ice-water particle size, shape, and mixing ratio $q_i$; aerosol properties; radiatively active gases; surface properties; and incoming solar radiation – is implicit:

$$Q(\lambda, \phi, t) = Q(N_d(\lambda, \phi, p, t), q_l(\lambda, \phi, p, t), f_c(\lambda, \phi, p, t)) \tag{2}$$

$$R(\lambda, \phi, t) = R(N_d(\lambda, \phi, p, t), q_l(\lambda, \phi, p, t), f_c(\lambda, \phi, p, t)) \tag{3}$$

Let $\boldsymbol{x}^A = \{N_d^A, \mathcal{L}^A, f_c^A\}$ and $\boldsymbol{x}^B = \{N_d^B, \mathcal{L}^B, f_c^B\}$ denote the cloud properties in runs $A$ and $B$. We then define *forward* and *backward* PRP as inserting one cloud property at a time from one run into the cloud field of the other and recalculating the radiative fluxes:

$$\delta_{A \to B} Q_\xi = Q(\{x_\xi^A, x_{\zeta \neq \xi}^B\}) - Q(\boldsymbol{x}^B) \tag{4}$$

$$\delta_{B \to A} Q_\xi = Q(\{x_\xi^B, x_{\zeta \neq \xi}^A\}) - Q(\boldsymbol{x}^A), \tag{5}$$

where $\delta$ denotes the difference in TOA flux when cloud property $\xi$ is substituted from run $A$ into run $B$ or run $B$ into run $A$, respectively, and $Q$ (or $R$, analogously) is recalculated using the offline version of the model's radiative transfer scheme. *Forward–backward* PRP is simply the average of the two, taking into consideration that reversing direction reverses the sign of the radiative-flux perturbation (e.g., Klocke et al., 2013):

$$\delta_{A \leftrightarrow B} Q_\xi = \frac{\delta_{A \to B} Q_\xi - \delta_{B \to A} Q_\xi}{2} \tag{6}$$

When $A$ denotes the preindustrial (PI)-emissions run and $B$ denotes the present-day (PD)-emissions run, the components of $\mathrm{ERF_{aci}}$ correspond to

$$F_{N_d} = \overline{\delta_{\mathrm{PI}\leftrightarrow\mathrm{PD}}Q_{N_d} + \delta_{\mathrm{PI}\leftrightarrow\mathrm{PD}}R_{N_d}} \tag{7}$$

$$F_{\mathcal{L}} = \overline{\delta_{\mathrm{PI}\leftrightarrow\mathrm{PD}}Q_{\mathcal{L}} + \delta_{\mathrm{PI}\leftrightarrow\mathrm{PD}}R_{\mathcal{L}}} \tag{8}$$

$$F_{fc} = \overline{\delta_{\mathrm{PI}\leftrightarrow\mathrm{PD}}Q_{fc} + \delta_{\mathrm{PI}\leftrightarrow\mathrm{PD}}R_{fc}}. \tag{9}$$

For other meanings of $A$ and $B$, as in the additional experiments performed in Sections 3.1–3.3, the equivalent expressions to Equations (7)–(9) describe pseudo-forcing components rather than forcing components; we denote them as $\tilde{F}_{N_d}$, $\tilde{F}_{\mathcal{L}}$, and $\tilde{F}_{fc}$.

In Equations (7)–(9),

$$\overline{\mathcal{F}} = \frac{1}{N}\sum_{i=1}^{N}\mathcal{F}(t_i) \tag{10}$$

indicates averaging over the time dimension of a field $\mathcal{F}$ evaluated at the $N$ time steps $\{t_1,\ldots,t_N\}$. "Evaluated" can, itself, refer to a temporal average over the interval between evaluation time steps, as in a 3-hourly or daily mean, or it can refer to the instantaneous value of the field at that time step; when the distinction matters (because $Q$ and $R$ are not linear functions of their input variables), we will indicate the averaging interval as $\overline{\mathcal{F}}^{(\Delta t)}$. Thus, $\overline{\mathcal{F}}^{(\mathrm{inst})}$ denotes the temporal mean of instantaneous model output, while $\overline{\mathcal{F}}^{(3\ \mathrm{h})}$ denotes the temporal mean of 3-hourly-averaged model output.

## 2.2 Model description

We use several model runs performed with the ECHAM–HAMMOZ model, version echam6.1–ham2.2–moz0.9 (Neubauer et al., 2014). The model is based on the ECHAM atmospheric general circulation model (Stevens et al., 2013), the HAM interactive aerosol module (Stier et al., 2005; Zhang et al., 2012), and the trace-gas chemistry module MOZ (Kinnison et al., 2007) (the latter is disabled in these runs). Of most direct relevance to our study, the parameterized processes contributing to warm-cloud–aerosol interactions are aerosol activation into cloud droplets according to Lin and Leaitch (1997); diagnostic warm rain processes (autoconversion and accretion) according to Khairoutdinov and Kogan (2000); and aerosol scavenging according to Croft et al. (2009, 2010). The stratiform cloud scheme is that of Lohmann and Roeckner (1996), extended to double-moment microphysics by Lohmann et al. (2007) and Lohmann and Hoose (2009), with the Sundqvist et al. (1989) critical-relative-humidity cloud cover parameterization.

To reduce internal variability and achieve low statistical uncertainty on the forcing components within a reasonable integration time, we use monthly varying but yearly repeating SST and SIC from the observed climatology and nudge the large-scale wind fields to the present-day ERA-Interim reanalysis (Dee et al., 2011) wind fields of the years 2000–2010 (in some sensitivity studies, only the year 2000 is used).

Estimates of radiative forcing are computed by performing a pair of model runs with present-day SST, SIC, and wind fields, and aerosol (precursor) emissions estimates for either the year 2000 or the year 1850. Emissions are from the AEROCOM-II ACCMIP dataset; in particular, anthropogenic emissions follow Lamarque et al. (2010).

To perform PRP on the model output, we have updated the offline version of the RRTM-based ECHAM6 radiative transfer code (Pincus and Stevens, 2013) that was originally used in Klocke et al. (2013). We use climatological aerosol mixing ratios (Kinne et al., 2013) rather than the model's time-varying aerosol fields for aerosol–radiation interactions; this reduces technical complexity as well as the volume of model output needed to perform the PRP calculation. To the extent that aerosol overlying cloud is a small effect, this choice mainly affects our estimate of the $f_c$ adjustment (Ghan, 2013; Zelinka et al., 2014), which, unlike the $N_d$ forcing and $\mathcal{L}$ adjustment, is straightforward to compute without the PRP machinery; comparison to Gryspeerdt et al. (2019b) shows that the $f_c$ adjustment estimate is not strongly affected by this simplification.

When clouds are absent (or the cloud fraction is very low) in one run and present in the other, perturbing cloud properties can yield unrealistically large or small $q_l$ or $N_d$ (and thus $r_e$); this "decorrelation" problem is well known from the application to climate feedbacks (Colman and McAvaney, 1997) in the context of the correlation between water vapor and cloudiness. We allow the radiative transfer code to resolve the conflicting cloud properties in the same way as it does when the cloud microphysics and cloud cover schemes produce conflicting cloud properties; in particular, $r_e$ can only vary within the limits of the cloud optics lookup table used by the model ($2 \times 10^{-6}$ to $32 \times 10^{-6}$ m). Appendices A1 and A2 describe tests we performed to verify that forward–backward PRP $\mathrm{ERF_{aci}}$ estimates are not impacted by the decorrelation problem.

## 3 Results

Since the components of the $\mathrm{ERF_{aci}}$ have not been diagnosed before in ECHAM–HAMMOZ by any method, we begin by presenting their global-mean values and geographic distributions in Section 3.1. In Section 3.2, we investigate whether rapid adjustments to the Twomey forcing are proportional to $F_{N_d}$ in terms of their spatial patterns. Section 3.3 investigates the sensitivity of the PRP results to the treatment of model columns containing ice and mixed-phase clouds. In Section 3.4, we determine how much temporal averaging is permissible before the PRP estimate becomes inaccurate. Sections A1 and A2 discuss whether PRP diagnoses the $\mathrm{ERF_{aci}}$ components correctly in the presence of decorrelation effects (i.e., effects of introducing one cloud property from one run into an uncorrelated cloud field in another run).

### 3.1 What are the $\mathrm{ERF_{aci}}$ components in ECHAM–HAMMOZ?

Using the PRP decomposition, Equations (7)–(9), we can diagnose the contributions to the $\mathrm{ERF_{aci}}$ from PD and PI-emission fixed-SST model runs. This is shown in Figure 1 and Table 1. The longwave effective forcing due to warm-cloud–aerosol interactions is small, as could be expected. No single forcing or adjustment dominates the shortwave $\mathrm{ERF_{aci}}$; the global-mean forcing $F_{N_d}$ and $\mathcal{L}$ adjustment $F_{\mathcal{L}}$ are of comparable magnitude at $-0.52 \ \mathrm{W\,m^{-2}}$ and $-0.53 \ \mathrm{W\,m^{-2}}$. The cloud-fraction adjustment $F_{f_c}$ at $-0.31 \ \mathrm{W\,m^{-2}}$ is the smallest of the components, consistent with CMIP5 models but not with other AeroCom models (Zelinka et al., 2014; Ghan et al., 2016; Gryspeerdt et al., 2019b). Our decomposition agrees with the results of Gryspeerdt et al. (2019b) to within $0.1 \ \mathrm{W\,m^{-2}}$ but disagrees with Ghan et al. (2016) (who estimate $F_{\mathcal{L}}/F_{N_d} \approx 5$, in contrast to our estimate of $\approx 1$), which both use a very similar configuration of the same version of ECHAM–HAMMOZ. Research to understand the differences between our decomposition and others is underway. All results in this section are based on 3-hourly

mean output, $\overline{F^{(3\,h)}}$. This is a commonly used model output configuration, albeit at the expensive end of the spectrum from the standpoint of storage space requirements. We will justify this choice in Section 3.4.

The geographic patterns of all components exhibit similar features. In the northern hemisphere, fairly strong forcing prevails over both oceans and over most of the continents, with the exception of desert regions, northern Asia, and the Arctic; over the continents, a plume of high forcing components over China, extending eastwards into the Pacific Ocean, and of a magnitude far greater than over Europe and North America, is especially pronounced. In the southern hemisphere, on the other hand, sizable forcing components are largely limited to the subtropical southern Pacific and southern Atlantic in the vicinity of the persistent stratocumulus decks; smaller local maxima in the forcing components also exist in the outflow regions of the midlatitude westerlies downwind of South America, Africa, and Australia.

These geographic patterns result from a convolution of the distributions of susceptible clouds and $N_d$ perturbations. Figure 2 shows the ERF$_{aci}$ sensitivity, defined as the forcing or adjustment strength per $e$-folding of the $N_d$ burden (with the bar denoting temporal averaging over the length of the run),

$$S_\xi = \frac{F_\xi}{\Delta \ln \overline{\mathcal{N}_d}} \tag{11}$$

where the $N_d$ burden is defined as

$$\mathcal{N}_d = \int N_d \, dz; \tag{12}$$

Equation (11) is similar to the factorization of Bellouin et al. (submitted). The sensitivity for all ERF$_{aci}$ components is greater over ocean than over land. Over land, the most susceptible clouds can be found over tropical Africa, southeastern Asia, and Central America; the anthropogenic $N_d$ perturbation over South America is too small to determine reliable sensitivities. Over ocean, the largest sensitivities occur in the midlatitudes, near the equator, and over the eastern ocean boundaries; again, $N_d$ perturbations in the southern hemisphere are too small to estimate sensitivities reliably. The regions of maximum sensitivity are the regions where the base-state $\mathcal{L}$ is high (not shown), which is intuitive in light of the strong influence of $\mathcal{L}$ on cloud albedo.

Figure 3 shows the anthropogenic perturbation in $\mathcal{N}_d$. The perturbation is generally stronger over the continents than over ocean; over ocean, it is generally strongest in the subtropics. Thus, the highest perturbations tend to coincide with the least susceptible clouds, which explains why the ERF$_{aci}$ components exhibit far less variability over a wide latitudinal range over the northern hemisphere oceans than either the sensitivity or the $N_d$ perturbation, and why the remote northern hemisphere oceans are comparable to the polluted continental regions in ERF$_{aci}$ strength. The exceptions to this pattern are the near-shore eastern ocean boundaries, where a high sensitivity and reasonably strong $N_d$ perturbation coincide, and eastern China, where a reasonably high sensitivity and very strong $N_d$ perturbation coincide. The location where ECHAM–HAMMOZ simulates both the strongest $N_d$ forcing and the strongest $\mathcal{L}$ adjustment is over land in China, extending downwind into the northwestern Pacific Ocean.

In observational studies or observationally constrained modeling studies, it is common to define susceptibilities analogously to Eq. (11) based on PD variability in cloud and aerosol variables and then multiply those susceptibilities by wholly or partially

(Bellouin et al., 2013; Kinne, 2019) model-derived estimates of anthropogenic aerosol perturbations. (In the terminology we adopt here, "sensitivity" is a change in cloud property or cloud radiative effect in response to a climatological change in an aerosol variable, whereas "susceptibility" is a change in response to an instantaneous change in an aerosol variable.) Our results show that the most susceptible oceanic clouds in this version of ECHAM–HAMMOZ occur where continental pollution intrudes on relatively clean conditions over the eastern ocean boundaries. Some observational (or observationally constrained) studies agree with this result (Quaas et al., 2008; Alterskjær et al., 2012; Engström et al., 2015; Gryspeerdt et al., 2016; Andersen et al., 2017; Christensen et al., 2017; McCoy et al., 2017), while others disagree (Lebsock et al., 2008; Chen et al., 2014; Gryspeerdt et al., 2017). Of those studies that are not restricted to oceanic clouds, some agree with our finding of strong forcing due to relatively susceptible clouds over China (Gryspeerdt et al., 2017; McCoy et al., 2017) and some do not (Quaas et al., 2008; Alterskjær et al., 2012; Gryspeerdt et al., 2016).

## 3.2 Are the adjustments proportional to the forcing?

An intriguing aspect of the ACI problem is whether the adjustments may be described approximately as a proportional response to the forcing (Gryspeerdt et al., 2019a). On the one hand, we do not necessarily expect proportionality in the physical atmosphere, since the processes responsible for the adjustments carry memory of the cloud evolution over various time scales; the parameterized cloud processes in GCMs share this feature, at least in principle, since the anthropogenic $N_d$ perturbation seen by the precipitation parameterization at one time step could be the result of a CCN perturbation at some point in the past, carried to another point in space by advection, and influenced by any of the other parameterized cloud processes. On the other hand, complex systems often exhibit simple emergent behaviors (e.g., Mülmenstädt and Feingold, 2018, and references therein). If the adjustments were to follow proportionally from the forcing, one consequence for the ACI problem would be that the total $ERF_{aci}$ uncertainty should not be estimated as the uncertainty on the sum of uncorrelated $RF_{aci}$ and adjustments but rather take the correlation between the forcing and adjustments into account, which would result in a smaller $ERF_{aci}$ uncertainty estimate.

In this study, we can test for proportionality in terms of the geographic distribution using the spatial variability in the temporal-mean $ERF_{aci}$ components. Figure 4 shows that the zonal mean of the ratio between $\mathcal{L}$ adjustment and $N_d$ forcing is relatively stable around unity between the southern and northern midlatitudes with fairly small interhemispheric differences except in the Southern Ocean. The picture is somewhat different for the ratio between $f_c$ adjustment and $N_d$ forcing, which is more latitudinally variable and more different between the northern and southern hemisphere. Figure 5 reinforces these conclusions, showing that $F_{\mathcal{L}}$ and $F_{N_d}$ are fairly tightly correlated with a regression relationship remarkably close to one-to-one, while the relationship between $F_{f_c}$ and $F_{N_d}$ is much looser.

One interpretation of these results is that the $ERF_{aci}$ components share a geographic pattern due to the fact that large effects result from the coincidence of large anthropogenic aerosol sources and susceptible clouds; the shared geographic pattern then leads to an approximately proportional relationship that breaks down farther from the source regions or where a different mixture of cloud processes dominates the cloud response (e.g., the Southern Ocean). The cloud cover scheme, which diagnoses $f_c$ from the grid mean relative humidity, to some extent decouples $f_c$ from the other cloud properties, which attenuates the influence of $N_d$ on $f_c$. Nevertheless, the vagueness of this argument is unsatisfactorily mismatched against the precision of the

$F_{\mathcal{L}}$–$F_{N_d}$ relationship, which suggests a deeper mechanism at play, e.g., that precipitation acts as a common sink process for both $N_d$ and $\mathcal{L}$.

Further evidence for proportionality comes from (Gryspeerdt et al., 2019b), who find an intermodel proportional relationship between global-mean forcing and rapid adjustments.

## 3.3 How should we treat columns containing ice?

In attempting to diagnose warm-cloud ACI forcing components, the question arises how ice-containing clouds should be handled. We can conduct the following set of experiments to determine the range of forcing strengths associated with different thermodynamic-phase treatments:

1. Perturb cloud properties in all cloudy model levels.

2. Perturb cloud properties in any liquid-containing cloudy model levels.

3. Perturb cloud properties in liquid-only cloudy model levels (default).

4. Perturb cloud properties in liquid-only cloudy columns.

5. Perturb cloud properties in liquid-only cloudy columns, correcting the result by their temporal occurrence fraction.

Table 2 summarizes the results. (For reasons of efficiency, we performed these sensitivity experiments on daily-mean output, shortwave flux only. We did not perform experiment 2 because we expect the result to lie between experiments 1 and 3, whose separation is already in the noise.) We conclude that how we choose to treat mixed-phase and ice clouds makes little difference in ECHAM–HAMMOZ, so long as we do not restrict ourselves to columns containing only warm clouds. In the latter case, correcting the forcing by the temporal occurrence fraction of liquid-only columns in each model latitude–longitude box approximately recovers the results when ice-containing columns are retained; however, there is some indication of diverging trends in $F_{N_d}$ (which decreases in magnitude as the ice filtering becomes more restrictive) and $F_{\mathcal{L}}$ (which increases in magnitude as the ice filtering becomes more restrictive). The ice-free column requirement is often made in passive remote sensing studies to prevent contamination from ice clouds overlying warm clouds and uncertainties in multilayer cloud retrievals.

## 3.4 Does temporal averaging bias the results?

As Table 3 shows, longer averaging periods underestimate the forcing, but the differences between instantaneous output (the model time step is 7.5 minutes, but we sample every 3 h to reduce the data volume) and 3 h averages is minimal. Multimodel ensembles which archive 3 h average output or 3 h subsampled instantaneous of column cloud properties, e.g., AeroCom and CFMIP2, are therefore amenable to treatment by the PRP method.

## 4 Conclusions

We have presented the first decomposition of the ACI effective forcing in ECHAM–HAMMOZ into a Twomey forcing component and rapid adjustments of $\mathcal{L}$ and $f_c$. In ECHAM–HAMMOZ, no single component dominates: $F_{N_d} = -0.52$ W m$^{-2}$, $F_{\mathcal{L}} = -0.53$ W m$^{-2}$, and $F_{f_c} = -0.31$ W m$^{-2}$; the Twomey forcing and $\mathcal{L}$ adjustment are approximately equally strong, and the $f_c$ adjustment is somewhat weaker, as in many other models. The global ERF is dominated by the northern-hemisphere forcing. Within the northern hemisphere, the strongest forcing components occur over land in China in $F_{N_d}$ and $F_{\mathcal{L}}$. As expected, the stratocumulus sheets over the eastern ocean basins also show strong responses in both hemispheres, as do the midlatitude North Atlantic and North Pacific.

The temporal-mean spatial patterns of $F_{N_d}$ and $F_{\mathcal{L}}$ are highly correlated, suggesting an effective proportionality in the $\mathcal{L}$ adjustment to the Twomey forcing even though the precipitation-suppression mechanism by which the $\mathcal{L}$ adjustment is parameterized in the model has inherent memory that could decouple it from the Twomey effect. The spatial patterns of the temporal-mean $F_{N_d}$ and $F_{\mathcal{L}}$, while sharing some of the same gross features, have a much less tight relationship than $F_{N_d}$ and $F_{\mathcal{L}}$.

In our study of ECHAM–HAMMOZ, the forcing components are fairly insensitive to how we treat columns containing both ice and liquid cloud condensate. Requiring that columns be free of ice and then correcting for the temporal fractional occurrence of ice cloud, a technique that is often necessary in observational studies, largely reproduces the results we obtain when we do not filter out such columns, albeit possibly causing an overestimate of the $\mathcal{L}$ adjustment and an underestimate of the $N_d$ forcing. (In interpreting the bearing of these results on analyses of satellite cloud retrievals, note that these studies do not necessarily apply the ice-free requirement at the coarse GCM scales of the present work, depending on whether they use gridded "level 3" data or the "level 2" native resolution of the retrieval algorithms.)

Through idealized sensitivity studies presented in the Appendix, we have showed that PRP is a viable method for accurately decomposing ERF$_{\text{aci}}$ into a $N_d$ forcing and $\mathcal{L}$ and $f_c$ adjustments. This is the case despite large artifacts that occur due to the decorrelated cloud property fields; the forward–backward technique advocated by Colman and McAvaney (1997) is capable of removing these artifacts.

PRP directly uses the intrinsic model knowledge of the time-varying, three-dimensional aerosol perturbation and resulting perturbation of cloud properties to diagnose the ERF$_{\text{aci}}$ components and their spatial patterns. This makes it a useful tool for providing context to other less resource-intensive decomposition methods (e.g., Ghan et al., 2016; Gryspeerdt et al., 2019b) or to intercomparison studies (e.g., Pincus et al., 2016; Smith et al., 2018) despite its demand for high-frequency vertically resolved model output.

## Appendix A: Validation of the PRP method for ACI decomposition

### A1  What is the effect of decorrelating the cloud properties?

Consider the results of forward and backward PRP plotted separately for the PD–PI experiment in Figure A1. Not only are the magnitudes grotesque, but taken at face value, they would imply a positive forcing in one direction and a negative forcing in the other. Furthermore, the spatial patterns bear no resemblance to that expected for $ERF_{aci}$. In this section, we investigate the consequences of these features for the $ERF_{aci}$ decomposition.

Any given atmospheric property is often correlated with many others. Substituting cloud properties one at a time breaks these correlations. For example, since ECHAM–HAMMOZ parameterizes precipitation suppression by aerosol, we expect a positive correlation between $N_d$ and $\mathcal{L}$ within a model run. If we substitute $\mathcal{L}$ from another run, the mechanistic link between $N_d$ and $\mathcal{L}$ through precipitation suppression, by which higher $N_d$ at a given point in time leads to higher $\mathcal{L}$ at later times, is broken, and, therefore, the correlation between $N_d$ and $\mathcal{L}$ is altered.

We estimate the strength of such decorrelation effects by performing two model runs with (almost exactly) the same model physics, both nudged to the same large-scale dynamics and with the same fixed SST; the only difference between the runs is that a parameter in the Khairoutdinov and Kogan (2000) formulation for the autoconversion rate, tuned for ECHAM–HAMMOZ,

$$Q_{\mathrm{aut}} = \gamma q_l^{\alpha} \left( \frac{N_d}{1\ \mathrm{cm}^{-3}} \right)^{-\beta}, \tag{A1}$$

has been slightly perturbed from $\beta = 1.79$ to $\beta' = 1.79 + 10^{-5}$ ($\alpha = 2.47$ and $\gamma = 4 \times 1350\ \mathrm{s}^{-1}$ are unchanged). Even over short integration times (a year), these model runs will converge on the same climate, with nearly identical forcing components. (The small perturbation in $\beta$ does not result in a significant change in model sensitivity.) However, at any given elapsed integration time and geographic location, the cloud properties in the two runs are essentially uncorrelated. We refer to this pair of runs as the *same-climate–different-weather experiment*. Knowing that the true climatological TOA flux difference between this pair of runs is zero, we can use these runs to estimate decorrelation effects between any other decorrelated pair of runs, including the PD and PI emissions runs.

We find that decorrelation effects cause the PRP method to estimate enormous TOA flux perturbations when we perform forward or backward substitution of any single cloud property; this is shown in panels (a) and (b) of Figure A2.

Unlike forward PRP or backward PRP individually, forward–backward PRP is unaffected by decorrelation, both in the global mean and locally in the temporal mean: panel (c) of Figure A2 shows that the fluctuations in $\Delta Q$ rapidly (i.e., within a year) average to zero. This confirms that the Colman and McAvaney (1997) prescription is successful at minimizing the spurious effects of decorrelation.

### A2  Does PRP give the right answer?

The preceding section provides evidence that strong decorrelation effects do not lead to a spurious offset in forward–backward PRP results. Next, we show that decorrelation effects also do not lead to spurious scale factors. To do so, we scale $N_d$ and $\mathcal{L}$ by a globally constant factor of 1.1 at all timesteps and scale $f_c$ by 0.99. We use PRP to diagnose the forcing associated with

each of these perturbations; the results are shown in the first three rows of Table A1. We can then estimate the strength of the decorrelation effects by performing PRP on the $\beta' = 1.79 + 10^{-5}$ run and the scaled-$\{N_d, \mathcal{L}, f_c\}$ $\beta = 1.79$ run. This is shown in the middle three rows of Table A1; the correct results are recovered to good approximation, with generally small attribution to incorrect $\mathrm{ERF_{aci}}$ components (the largest is $-0.05\ \mathrm{W\,m^{-2}}$ incorrectly diagnosed as $f_c$ adjustment in the $\mathcal{L} \times 1.1$ experiment) and

generally small differences between the actual and diagnosed (the largest is a diagnosed $F_{\mathcal{L}} = -0.48\ \mathrm{W\,m^{-2}}$ when the correct value is $-0.53\ \mathrm{W\,m^{-2}}$). The final test is an experiment in which all cloud properties are perturbed simultaneously and the clouds are decorrelated by using a $\beta' = 1.79 + 10^{-5}$ baseline run. The results are shown on the last line of Table A1; the correct $\mathrm{ERF_{aci}}$ components are recovered in the presence of the confounding effects of decorrelation and of perturbing multiple cloud properties simultaneously with $0.1\ \mathrm{W\,m^{-2}}$ or better accuracy, the largest discrepancy being the diagnosed $F_{f_c} = 0.14\ \mathrm{W\,m^{-2}}$

when the correct value is $0.24\ \mathrm{W\,m^{-2}}$).

     Thus, we find that forward–backward PRP can diagnose the forcing components correctly in the presence of decorrelations, in addition to diagnosing the absence of forcing correctly in the same-climate–different-weather case.

## A3    Are the results sensitive to choosing grid-mean or in-cloud perturbations?

Perturbing in-cloud or grid-mean $N_d$ and $\mathcal{L}$ would be equivalent in the limit in which TOA flux perturbations are linear in the

cloud properties. While individual model columns do not satisfy this linearity requirement, the temporal mean apparently exhibits sufficient effective linearity that the choice of in-cloud or grid-mean perturbations has little effect on the ERF component estimate; compare Tables 1 and A2.

*Code and data availability.* The PRP code is available in the GitHub repository https://github.com/jmuelmen/acp-2018-1304-echam-prp, archived under source-code DOI https://doi.org/10.5281/zenodo.3457389. The analysis code is available in the GitHub repository https:

20     //github.com/jmuelmen/acp-2018-1304, archived under source-code DOI https://doi.org/10.5281/zenodo.3457496. ECHAM–HAMMOZ is available from https://hammoz.ethz.ch subject to acknowledgment of a licensing agreement. Due to the large data volume of 3-hourly vertically resolved fields, the model output itself was not archived, but model configuration files that can be used to replicate the output are available as part of the PRP code. The PRP output on which the manuscript is based is available from the data DOI https://doi.org/10.5281/zenodo.3457397.

*Author contributions.* $j_\mu$ designed the study, performed model runs, computed the PRP diagnostics, and drafted the manuscript. EG and $j_\mu$ validated and debugged the results. EG, MS, PM, JQ, and $j_\mu$ proposed various sensitivity studies. EG, MS, PM, SD, JQ, and $j_\mu$ worked on the interpretation of the results. All authors contributed text to or comments on the manuscript.

*Competing interests.* The authors declare that they have no competing interests.

*Acknowledgements.* We thank Christina Sackmann, Steve Ghan, two anonymous reviewers, and the editor for their insightful comments that have improved the manuscript. The ECHAM–HAMMOZ model is developed by a consortium composed of ETH Zurich, Max-Planck-Institut für Meteorologie, Forschungszentrum Jülich, University of Oxford, and the Finnish Meteorological Institute and managed by the Center for Climate Systems Modeling (C2SM) at ETH Zurich. This work was funded by the FLASH project (project number QU 311/14-1) in the HALO Priority Program (SPP 1294) of the German Research Foundation (Deutsche Forschungsgemeinschaft, DFG); by the European Union through European Research Council (ERC) Starting Grant QUAERERE (grant agreement 306284); by the German Federal Ministry of Education and Research within the Research for Sustainable Development (FONA) framework program under the HD(CP)$^2$ project (project number 01LK1504C). EG was supported by an Imperial College London Junior Research Fellowship. PM's contributions were supported by a travel grant from the Leibniz Invitations program at Universität Leipzig. Computing resources were provided by the German Climate Computing Center (Deutsches Klimarechenzentrum, DKRZ). We acknowledge support from the Open Access Publishing program of DFG and Universität Leipzig.

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

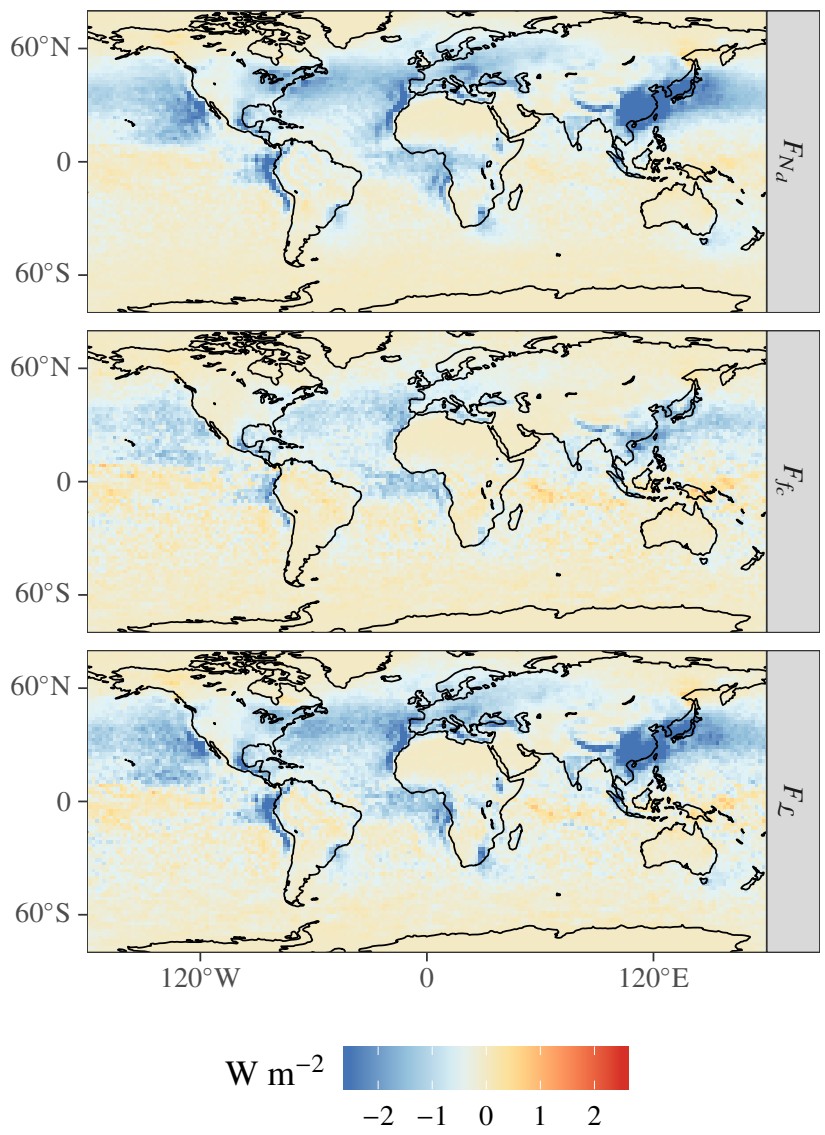

**Figure 1.** ERF$_{aci}$ components in ECHAM–HAMMOZ estimated by forward–backward PRP

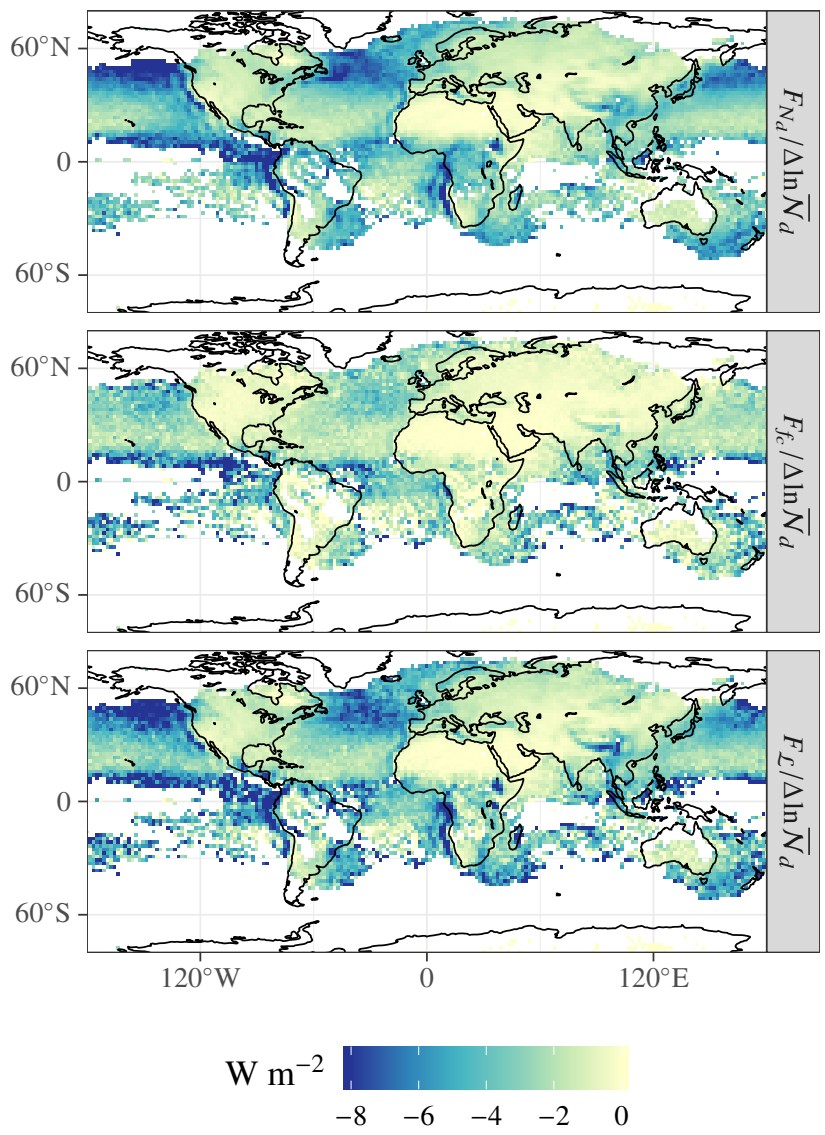

**Figure 2.** Sensitivity of ERF$_{aci}$ components to anthropogenic $\mathcal{N}_d$ change (shown only where $\Delta \ln \overline{\mathcal{N}}_d > 0.05$)

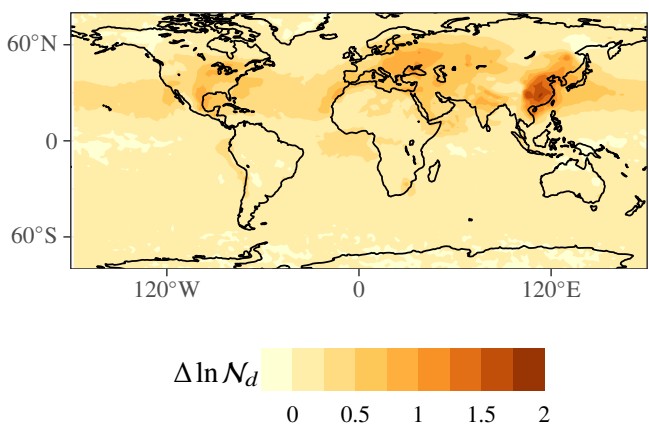

**Figure 3.** Anthropogenic $\ln \mathcal{N}_d$ change

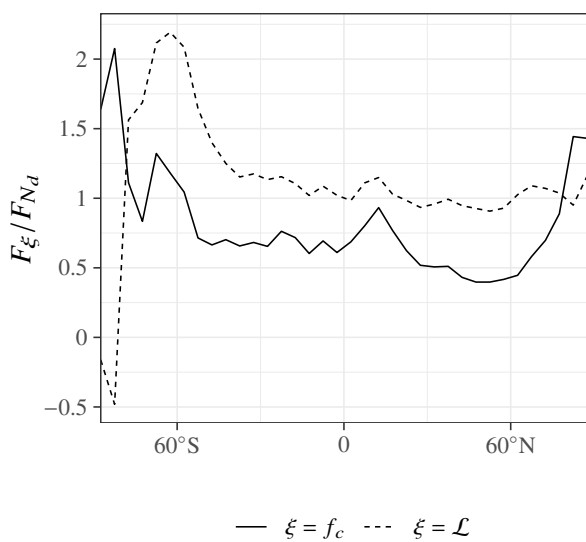

**Figure 4.** ERF$_{\text{aci}}$ adjustments relative to the Twomey forcing

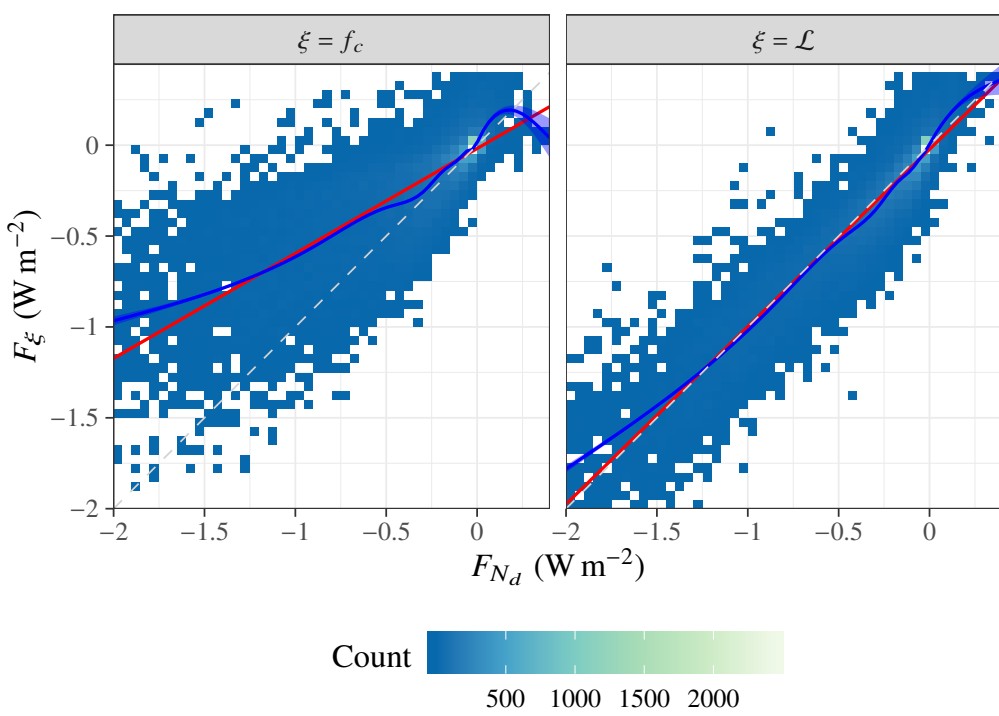

**Figure 5.** Correlation plots between the temporal-mean Twomey forcing and the adjustments; color indicates the number of grid boxes within each $0.05\,\mathrm{W\,m^{-2}} \times 0.05\,\mathrm{W\,m^{-2}}$ bin; the red line is a linear least-squares regression; the blue line is a generalized additive model regression (Wood, 2011), with 95% confidence interval shaded in light blue; and the dashed gray line is the one-to-one line

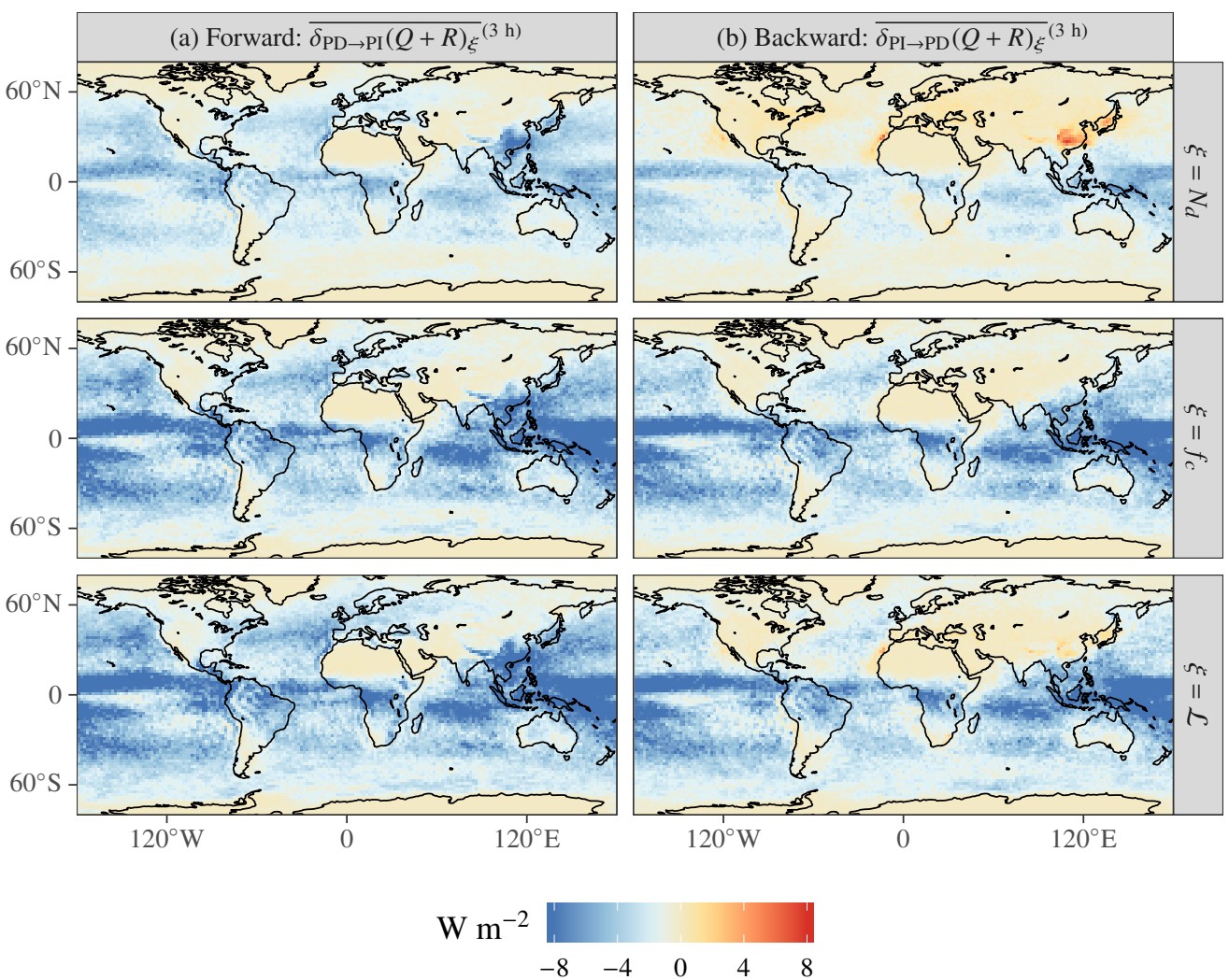

**Figure A1.** Forward (a) and backward (b) PRP estimates of the ERF$_{\text{aci}}$ components. Note the significantly wider color scale than in Figure 1.

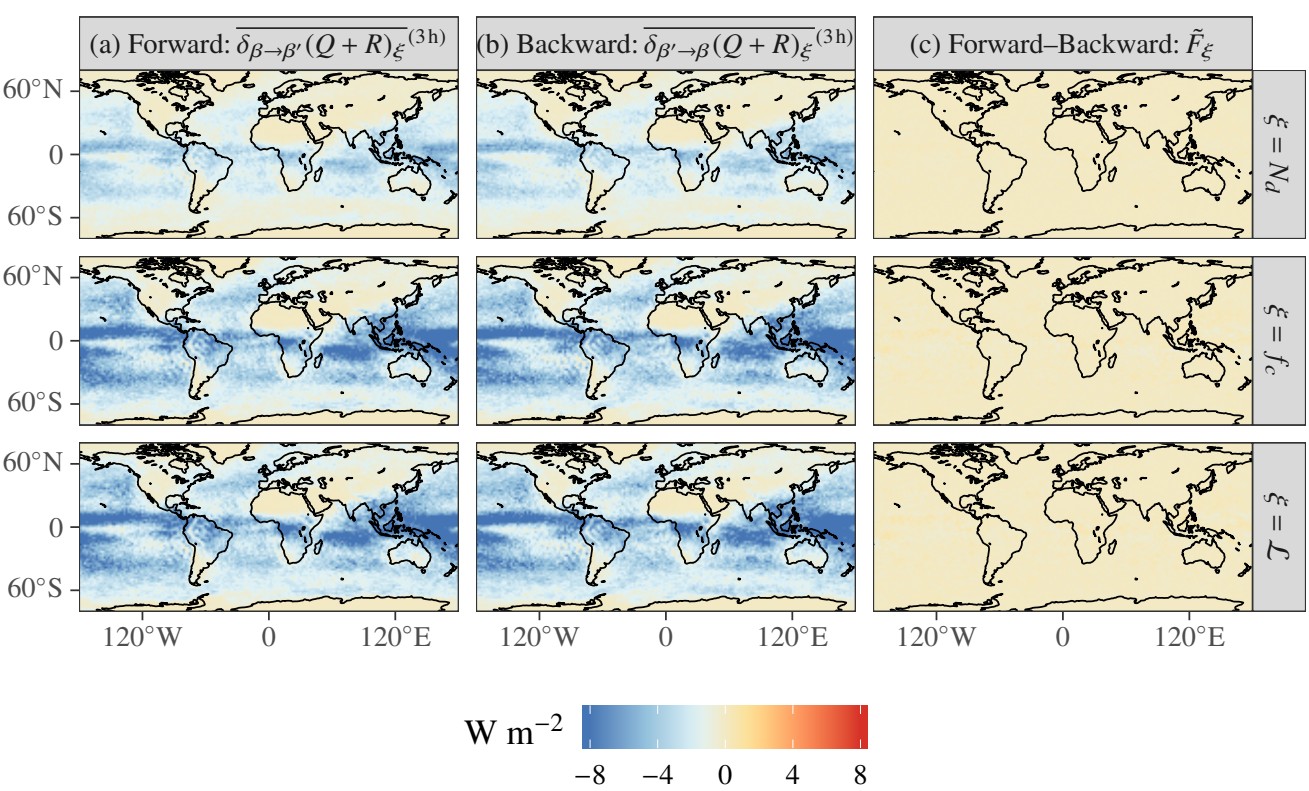

**Figure A2.** Forward (a), backward (b), and (c) forward–backward PRP performed on the same-climate–different-weather case. Note the significantly wider color scale than in Figure 1.

**Table 1.** ERF$_{aci}$ components in ECHAM–HAMMOZ estimated by forward–backward PRP. The total ERF also includes the ice-phase ACI effects (−0.59 W m$^{-2}$ in the SW, 0.88 W m$^{-2}$ in the LW), RF$_{ari}$ (−0.17 W m$^{-2}$ in the SW), and a negligible surface-albedo contribution (−0.01 W m$^{-2}$), estimated for a very similar model run in Gryspeerdt et al. (2019b). The sum of the components thus balances at approximately the 0.2 W m$^{-2}$ level, a relative error similar to the 0.1 W m$^{-2}$ estimated uncertainty on the ERF$_{aci}$ components.

| Spectrum | ERF components (W m$^{-2}$) | | | Sum (W m$^{-2}$) | Total ERF (W m$^{-2}$) |
|---|---|---|---|---|---|
| | $F_{N_d}$ | $F_{f_c}$ | $F_{\mathcal{L}}$ | $F_{N_d} + F_{f_c} + F_{\mathcal{L}}$ | |
| LW | 0.00 | 0.04 | 0.03 | 0.07 | 0.72 |
| SW | −0.52 | −0.35 | −0.57 | −1.44 | −2.03 |

**Table 2.** Dependence of diagnosed $ERF_{aci}$ components on treatment of thermodynamic phase

| Phase treatment | ERF components (W m$^{-2}$) | | |
|---|---|---|---|
| | $\delta Q_{N_d}$ | $\delta Q_{f_c}$ | $\delta Q_{\mathcal{L}}$ |
| All phases | −0.29 | −0.29 | −0.34 |
| Liquid-only cloudy model levels | −0.27 | −0.27 | −0.35 |
| Liquid-only cloudy model columns | −0.15 | −0.17 | −0.21 |
| Liquid-only cloudy model columns (corrected for occurrence fraction) | −0.26 | −0.29 | −0.38 |

**Table 3.** Dependence of diagnosed $ERF_{aci}$ components on temporal averaging

| Averaging period | ERF components (W m$^{-2}$) | | |
|:---:|:---:|:---:|:---:|
| | $F_{N_d}$ | $F_{f_c}$ | $F_{\mathcal{L}}$ |
| 1 month | −0.09 | −0.09 | −0.11 |
| 1 d | −0.35 | −0.33 | −0.30 |
| 3 h | −0.52 | −0.31 | −0.53 |
| instantaneous | −0.55 | −0.30 | −0.51 |

**Table A1.** ERF$_{\text{aci}}$ components resulting from idealized perturbations to $N_d$, $\mathcal{L}$, and $f_c$; estimate of the same ERF$_{\text{aci}}$ components by forward–backward PRP in the presence of decorrelation effects.

| Perturbation | ERF components (W m$^{-2}$) | | |
|:---:|:---:|:---:|:---:|
| | $F_{N_d}$ | $F_{f_c}$ | $F_{\mathcal{L}}$ |
| $N_d \times 1.1$ | −0.38 | −0.00 | −0.00 |
| $f_c \times 0.99$ | −0.00 | 0.24 | −0.00 |
| $\mathcal{L} \times 1.1$ | −0.00 | −0.00 | −0.53 |
| $N_d \times 1.1$ with $\beta' = \beta + 10^{-5}$ | −0.37 | −0.01 | −0.01 |
| $f_c \times 0.99$ with $\beta' = \beta + 10^{-5}$ | 0.01 | 0.21 | 0.02 |
| $\mathcal{L} \times 1.1$ with $\beta' = \beta + 10^{-5}$ | 0.01 | −0.05 | −0.48 |
| $N_d \times 1.1, \mathcal{L} \times 1.1, f_c \times 0.99$ with $\beta' = \beta + 10^{-5}$ | −0.31 | 0.14 | −0.49 |

**Table A2.** ERF$_{aci}$ components calculated by PRP on $f_c$ and in-cloud $N_d$ and $q_l$

| ERF components (W m$^{-2}$) | | |
|---|---|---|
| $F_{N_d}$ | $F_{f_c}$ | $F_{\mathcal{L}}$ |
| −0.48 | −0.30 | −0.48 |