# Peer review of "Separating radiative forcing by aerosol–cloud interactions and rapid cloud adjustments in the ECHAM–HAMMOZ aerosol–climate model using the method of partial radiative perturbations"

_Atmospheric Chemistry and Physics, 2018_

## Referee Comment (RC1) · Ghan (Referee) · 23 Jan 2019

This study proposes to use the partial radiative perturbation method to separate contributions of changes in droplet number, liquid water content, and cloud fraction to estimates of effective radiative forcing by aerosol-cloud interactions. The method is described fairly well, except as noted. The authors present results that show conclusively that the method is not biased. The title captures the method and conclusions. The

abstract succinctly and clearly summarizes the results. The referencing is adequate, except as noted.

page 1, line 4. Insert "by anthropogenic cloud droplet number change" after "radiative forcing".

Page 3, lines 15-16. Consider the decomposition expressed by equations 6-8 in Ghan et al. PNAS 2016.

Page 4. I'm concerned about substituting a cloud property from one run into diagnostic radiation calculations from another run, since cloud properties vary in time. What is done when clouds at a particular time are simulated in one run but not in the other. How is the cloud property determined then? Use time mean property will work if cloud forms at least once at that point, but what if it never forms at that point in one simulation but does in the other? This issue is mentioned later: large artifacts that occur due to the decorrelated cloud property fields, and tested in the Appendix, but it does not address the question of how to specify properties of clouds not present in one simulation.

Page 6, line 7. Insert "global mean" before "forcing".

Page 9, lines 29-32. Is PRP the most direct method? Is it more direct than the method described by Ghan et al. PNAS 2016? Why not compare the two methods? The Ghan method is simple to implement.

Figure 2 Caption has a question.

ACPD

---

## Referee Comment (RC2) · Anonymous Referee #2 · 14 Feb 2019

The Partial Radiative Perturbation approach is a common offline approach for diagnosing forcing terms. Traditionally it has been used to decompose non-cloud terms. For the first time, this note uses PRP to decompose aerosol cloud forcing terms, diagnosing forcing and adjustments due to changes in droplet number, liquid water path and cloud fraction. They find RFaci and liquid water path adjustments are similar in magnitude and highly correlated while cloud fraction adjustments are smaller in magnitude and less well correlated to RFaci. The note is well written, novel and appropriate for
full publication if the few comments below can be adequately addressed:

Page 2, line 17: In RFari you are using the ARI abbreviation for the first time. You should more clearly spell out what this abbreviation means (even though you mention the phrase "aerosol-radiation" in the line above).

Page 3, Line 1: On a similar note, please formally define fc as cloud fraction. I don't see it defined anywhere.

Page 3, line 17: "... but this decomposition does not correspond to the forcing-andadjustment decomposition." More or clearer explanation about why APRP does not fit the forcing-adjustment framework would be helpful. This was a bit vague.

Page 6, line 20. A specific example reference of the observational studies you talk about would be helpful here.

Table 1: Does RFari account perfectly for the difference between the sum of the ERFaci and the total ERF? Or is there some error associated with the PRP method in that difference? It would be good to quantify RFari. Perhaps with double-call calculations or the Ghan method.

Figure 1: Any explanation for the local maximum in forcing/adjustment terms along the eastern boundary currents? Right along the west coast of N. America, S. America and Europe? It seems these are also regions where the backwards and forwards PRP calculations differ notably (Figure A1)

Figure 2: The caption seems to include an editing note by accident.

A3: I'd prefer the appendix discussion and figure about temporal averaging to be included in the main section of the note, especially since it is given a prominent spot in the abstract. Given the recent push for large model comparison projects to include forcing diagnosis (where temporally averaged data is the norm), this result seems important.

2019.

---

## Author Comment (AC1) · 10 Jun 2019

We thank the reviewer for his or her thorough reading of the manuscript and helpful comments. Please find our responses inline below.

*Page 2, line 17: In RFari you are using the ARI abbreviation for the first time. You should more clearly spell out what this abbreviation means (even though you mention the phrase "aerosol–radiation" in the line above).*

[Figure]

Thank you for pointing out this omission. We have defined the abbreviation in the revised manuscript.

*Page 3, Line 1: On a similar note, please formally define $f_c$ as cloud fraction. I don't see it defined anywhere.*

Thank you for pointing out this omission, as well. We have defined the variable in the revised manuscript.

*Page 3, line 17: "... but this decomposition does not correspond to the forcing-and-adjustment decomposition." More or clearer explanation about why APRP does not fit the forcing-adjustment framework would be helpful. This was a bit vague.*

We thank the reviewer for pointing out the vague language. In the revised manuscript, we now explain that APRP decomposes the cloud property changes into changes in area fraction, cloud albedo, and cloud absorption. The change in area fraction maps well onto the cloud fraction adjustment in the forcing–adjustment framework, but the APRP cloud albedo change includes both the effect of the anthropogenic $N_d$ change and the $\mathcal{L}$ adjustment.

*Page 6, line 20. A specific example reference of the observational studies you talk about would be helpful here.*

We have expanded the discussion in this paragraph, also in light of the reviewer's comment on Fig. 1 below. We now separately discuss changes over ocean and over land and compare the patterns we have derived for each to a number of observational or observationally constrained modeling studies. We have also factorized the geographic distributions of the ERFaci components into an anthropogenic $N_d$ perturbation and a model sensitivity to the perturbation, to facilitate comparison to observational estimates of aerosol susceptibilities.

*Table 1: Does RFari account perfectly for the difference between the sum of the ERFaci and the total ERF? Or is there some error associated with the PRP method in that*

*difference? It would be good to quantify RFari. Perhaps with double-call calculations or the Ghan method.*

Very good point; the RFari and ice-cloud ACI are not part of our decomposition, but they are estimated for a very similar model run in Gryspeerdt et al. (submitted; should be in ACPD by the time this manuscript is published). The total ERF also includes the ice-phase ACI effects ($-0.59$ W m$^{-2}$ in the SW, $0.88$ W m$^{-2}$ in the LW), RFari ($-0.17$ W m$^{-2}$ in the SW), and a negligible surface-albedo contribution ($-0.01$ W m$^{-2}$). The sum of the components thus balances at approximately the 0.2 W m$^{-2}$ level, a relative error similar to the 0.1 W m$^{-2}$ estimated uncertainty on the ERFaci components.

*Figure 1: Any explanation for the local maximum in forcing/adjustment terms along the eastern boundary currents? Right along the west coast of N. America, S. America and Europe? It seems these are also regions where the backwards and forwards PRP calculations differ notably (Figure A1)*

We have expanded the discussion in Sec. 3.1 to better describe and explain these local maxima: these are regions where low clouds are abundant ($\mathcal{L}$ is large) and anthropogenic aerosols mix from the continents into the cleaner maritime air masses.

Regarding the last sentence of the comment, the forcing estimate, by construction, is highest where the differences between forward and backward PRP are greatest.

*Figure 2: The caption seems to include an editing note by accident.*

Thank you for pointing out this leftover editing detritus. We have removed it from the manuscript.

*A3: I'd prefer the appendix discussion and figure about temporal averaging to be included in the main section of the note, especially since it is given a prominent spot in the abstract. Given the recent push for large model comparison projects to include forcing diagnosis (where temporally averaged data is the norm), this result seems important.*

We agree; it is a bit strange to have to consult the appendix for one of the main points

of the paper. We have moved Appendix 3 into the results section (Sec. 3).

---

## Author Comment (AC2) · 10 Jun 2019

We thank Steve Ghan for his thorough reading of the manuscript and helpful comments. Please find our responses inline below.

*Page 1, line 4. Insert "by anthropogenic cloud droplet number change" after "radiative forcing".*

[Figure]

Thank you for suggesting this clarification. We have adopted the change in the revised manuscript.

*Page 3, lines 15–16. Consider the decomposition expressed by equations 6–8 in Ghan et al. PNAS 2016.*

Thank you for pointing out the imprecise wording in this sentence. Our intended meaning was to differentiate between, on the one hand, "exact" methods that use the time-varying, three-dimensional state of the model (i.e., the model's direct knowledge of the anthropogenic perturbations); and, on the other hand, methods that require idealizing the cloud as globally homogeneous or performing statistical analysis such as linear regression on the model output. We have noted this in the revised manuscript. We have also changed "exact" to "direct", since we show later on that our method still carries uncertainties on the order of $0.1\ \mathrm{W\ m^{-2}}$.

*Page 4. I'm concerned about substituting a cloud property from one run into diagnostic radiation calculations from another run, since cloud properties vary in time. What is done when clouds at a particular time are simulated in one run but not in the other. How is the cloud property determined then? Using time mean property will work if cloud forms at least once at that point, but what if it never forms at that point in one simulation but does in the other? This issue is mentioned later: large artifacts that occur due to the decorrelated cloud property fields, and tested in the Appendix, but it does not address the question of how to specify properties of clouds not present in one simulation.*

We were concerned, too, and we suspect this problem has dissuaded others from trying PRP earlier. When clouds are absent in one run and present in the other, we let the radiative transfer resolve the conflicting cloud properties in the same way as it does when the cloud microphysics and cloud cover schemes produce conflicting cloud properties, i.e., effective radii can only vary within the limits of the cloud optics lookup table. This is guaranteed to produce incorrect results for the model column in which the

mismatch of cloud properties occurs; in fact, the correct result is probably undefined. However, we would consider this to be the heart (or perhaps the logical extreme) of the decorrelation problem. This is the reason we designed the tests in Sec. A2, where the correct forcing components are known, and found that the forward–backward PRP results agree with the correct values to within 0.1 W m$^{-2}$ accuracy, as you point out in your comment.

We agree that the issue of cloud presence in one run and absence in the other should be discussed in the text, and that the discussion should include a prescription for what to do when this case occurs. We have expanded the revised manuscript accordingly.

*Page 6, line 7. Insert "global mean" before "forcing".*

Thank you for suggesting this clarification. We have adopted the change in the revised manuscript.

*Page 9, lines 29–32. Is PRP the most direct method? Is it more direct than the method described by Ghan et al. PNAS 2016? Why not compare the two methods? The Ghan method is simple to implement.*

Thank you for pointing out the imprecise wording in this sentence. "Direct" was meant in the same way that "exact" was meant on p. 3, l. 15–16; we have clarified this in the revised manuscript. The suggestion of an intercomparison of methods is a good one, in particular as several additional decomposition methods are close to publication (Gryspeerdt et al., submitted, and at least one other study, private comm.). In the revised manuscript, we mention that the Ghan et al. (2016) estimate of $F_{\mathcal{L}}/F_{N_d} \approx 5$ is much greater than our result of $\approx 1$. However, we feel that tracking down the sources of differences between methods is best left for a dedicated intercomparison study.

*Figure 2 Caption has a question.*

Thank you for pointing out this leftover editing detritus. We have removed it from the manuscript.

---

## Author Response (AR2)

Fakultät für Physik und
Geowissenschaften
**Institut für Meteorologie**
Johannes Mülmenstädt

Universität Leipzig, Institut für Meteorologie, PF 232101, 04081 Leipzig

September 22, 2019

Dear Xiahong,

thank you for reading the manuscript closely and suggesting improvements. Please find a marked-up version of the manuscript showing the changes we made in response to your suggestions; consistently referring to the model as ECHAM–HAMMOZ; and adding data and source-code DOIs to the code availability section.

Best regards,

Johannes (for the authors)

[revised manuscript text omitted]